# Examining recovery experiences as a mediator between physical activity and study-related stress and well-being during prolonged exam preparation at university

**Tom Reschke** [1]*, **Thomas Lobinger**[1], **Katharina Reschke**[2]

**1** Institute for Civil Law, Labor and Commercial Law, Heidelberg University, Heidelberg, Germany, **2** Institute for Education Studies, Heidelberg University, Heidelberg, Germany

* tom.reschke@jurs.uni-heidelberg.de

**Data Availability Statement:** The data are now held in figshare and published with the following DOI: 10.6084/m9.figshare.25659129.

## Abstract

Prolonged study periods such as preparing for comprehensive exams pose a significant source of chronic stress for university students. According to the Conservation of Resources Theory, the replenishment of resources during leisure time is essential to a successful stress recovery process. This study examined the role of recovery experiences as a mediator of the relationship between physical activity as one specific recovery activity and both study-related stress and well-being. We applied a longitudinal design and approached students on three measurement occasions over seven months. The sample consisted of $N = 56$ advanced law students enrolled at one large German university who were all in their exam preparation to take their final exams. Students gave self-reports on their respective levels of physical activity (predictor), and recovery experiences (mediator), as well as on their study-related stress and well-being (outcomes). Results showed a negative trend in recovery-related variables and the outcomes as exam preparation progressed. There were mostly small correlations between physical activity and both stress and well-being at each measurement occasion. Recovery experiences partially mediated the relationship between physical activity and the outcomes on some measurement occasions. Our results suggest that the positive effects of recovery experiences related to physical activity become more sustained as exam preparation progresses and have a particularly positive impact on well-being. Future research can build on these findings by further examining recovery as an important means to help students better cope with long-lasting and stressful study periods.

## 1. Introduction

Navigating through university life requires students to overcome various challenges and taking academic examinations has been shown to be one of the most important stressors [1, 2]. During exam periods, students not only report elevated levels of stress [3, 4] and reduced levels of well-being [e.g., 1], but also engage in less health-related behaviors such as reduced physical

**Funding:** This work was funded by the Federal Ministry of Education and Research (BMBF) and Baden-Wuerttemberg under the Excellence Strategy of the Federal Government and the Länder. Form number: 2302096 Project number: D.801000/18.005.

**Competing interests:** The authors have declared that no competing interests exist.

activity [5, 6]. Study-related stress during exam periods can have both acute and chronic characteristics and relies on invested time for preparation (e.g., several weeks vs. months) and the relative importance of the exam and its grading. Many studies have argued that students are confronted with more pronounced academic stressors in state examination formats such as in medical school [7, 8]. This is because state exams typically require students to prepare extensive study material over several weeks and even months.

The German system of legal education at university-level is a particular example of a state examination format that requires students to undergo a long preparation time to finish their formal studies and pass their first state examination [9, 10]. Exam preparation takes about 18 months for most students and it is necessary to study an extensive amount of material and to take mock exams to practice solving legal cases. Most students reduce their participation in stress-relieving activities that have the potential to improve well-being (i.e., physical activity) to accomplish their academic objectives. Therefore, exam preparation constitutes an intense study period that increases the likelihood of chronic stress which has been linked to adverse psychological and physical health outcomes [3, 11]. Although acute stress has positive effects such as providing individuals with energy to master academic challenges, chronic stress affects academic performance [12] and is directly linked to mental health problems in students [13–15]. For example, chronic stress has repeatedly been shown to be a risk factor for developing depression and anxiety disorders [16, 17].

In a recent work, Giglberger et al. conducted a multi-method longitudinal study and followed 452 advanced law students over a 13-month period of their exam preparation [3]. This exam group was studied along with a control group of students that did not start with their exam preparation. Over time, there was a significant difference in perceived stress levels between both groups with exam students showing a continuous increase until the exam whereas controls were stable in their stress levels. For exam students, this pattern could also be observed for symptoms of anxiety and depression as well as several facets of perceived chronic stress. Those results were also underpinned by psychobiological markers. The study concluded exam preparation poses a significant source of stress for students both on a psychological and physiological level but remained unrevealing about practical implications [3]. However, chronically elevated stress levels and the deterioration of overall functioning call for stress recovery to avoid serious health problems.

## 1.1 Theoretical overview of recovery research

Stress is defined as the subjective discrepancy between external or internal demands and available coping resources in a given situation that an individual usually perceives as unpredictable and uncontrollable [18]. Well-being includes two components: subjective cognitive evaluations of a person's life and important domains (satisfaction) as well as affective states (positive and negative affect) [19]. Continuous and prolonged exposure to stressors makes recovery an important counterpart to feeling stressed and not well. Lack of recovery after stressful periods has been identified as a key factor in explaining stress-related health problems in occupational contexts [e.g., 20, 21]. Recovery describes the unwinding and restoration processes that lead a person's stress level to return to pre-stressor levels if it has previously increased in response to a particular demand [21]. Importantly, recovery is an everyday phenomenon that can be initiated and experienced during various occasions such as work breaks and during leisure time (e.g., free evenings or weekends).

The Conservation of Resources Theory (COR) [22, 23] is one of the predominant theoretical approaches for understanding the recovery process. Within this framework, work is assumed to be effortful, resources are depleted to meet demands, and the process of recovery is

required to replenish these resources. Therefore, time spent with recovery activities and experiencing recovery as such should help individuals restore resources. COR theory regards resources to play a key role in stress and recovery processes since resources are valuable means helping us to achieve goals which makes it important to protect and conserve them while threats to these resources or even loss can lead to stress-related outcomes [22–24]. According to COR theory, resources may be restored by investing additional resources such as engaging in certain activities to initiate and sustain the recovery process. To explore what promotes recovery, researchers have identified two distinct factors that are closely related to each other: recovery activities and recovery experiences. Both are considered the two driving mechanisms of recovery.

## 1.2 The role of recovery activities

Recovery activities (or behaviors) refer to what individuals do during work breaks or leisure time. In line with COR theory, certain activities such as exercising or meeting friends are not only assumed to provide resources and promote recovery rebuilding resources, but to help diminish the physiological stress response (i.e., resource-providing activities) [25]. For example, resource-providing recovery activities such as physical exercise and social activities have been shown to be positively associated with well-being and feelings of recovery [26]. To achieve recovery from work, sport and exercise belong to the most effective recovery activities [26]. Many studies have established a significant link between physical activity and both physical health (e.g., cardiovascular diseases, diabetes, sleep) and psychological health (e.g., well-being, stress, psychopathological symptoms) in work-related contexts [27, 28].

For the situation of university students, empirical evidence is less elaborated but paints a similarly positive picture in favor of physical activity. Studies have underlined the moderately positive association between physical exercise and mental health outcomes [29–31]. Final examinations have also received attention from a few researchers to determine the relationship between study-related stress and students' levels of physical activity. Two studies examined students both at the beginning of a semester and during an exam period and also included a control group that was assessed at the end of the semester without taking final exams [5, 32]. While both studies found reductions in the duration of physical activity compared to controls, Oaten and Cheng also discovered reductions in exercise frequency [5]. Griffin et al. demonstrated that those students who experienced higher study-related demands during stressful exam periods were also less likely to engage in physical activity [33]. There is a lack of empirical findings on prolonged study periods such as exam preparation that lasts over several months and puts students at risk for developing stress-related health problems.

## 1.3 The role of recovery experiences

Recovery experiences refer to what individuals perceive during and after recovery activities and have been divided into four core psychological states: psychological detachment from work, relaxation, mastery, and control [34]. Psychological detachment can be understood as an individual's experience of being mentally disconnected from work during free time. Relaxation refers to the experience of low sympathetic activation that can occur when engaging in calming activities during leisure time. Mastery goes along with an individual's experience of being challenged outside the work context which can occur when participating in activities that promote learning and personal growth. Control refers to an individual's need for self-determination during free time such as choosing certain activities on one's own terms. All four psychological states form a combined concept of recovery experiences and have been found to show low to moderate positive intercorrelations [34, 35]. Studies have provided ample

evidence that all four recovery experiences are negatively related to stress-related health complaints, exhaustion, and depression and positively related to psychological and psychosomatic well-being as well as performance [20, 34].

Recovery is considered a key mechanism by which individuals can enhance their overall well-being with recovery being equally related to psychosomatic outcomes and psychological outcomes [20]. For example, recovering from work has been shown to have significant positive effects on mind and body, as shown by the links with sleep and state positive and negative affect (short-term) as well as with life satisfaction (long-term effects) [20]. Importantly, however, psychological recovery has mostly been studied in samples from the workplace. There is good reason to presume that the same recovery mechanisms apply to other populations such as students in an academic context as well. University students face similar stressors to those experienced at work and academic stress can be compared to occupational stress in many respects.

### 1.4 Recovery experiences as mediators

It is reasonable to assume that recovery activities have positive effects on health-related outcomes by acting as necessary precursors to recovery experiences. Recovery experiences can be seen as the actual mechanism that contribute to the restoration of resources. In line with COR theory, recovery experiences transmit the effects of leisure activities into resource replenishment [22]. Recovery activities such as physical exercise should lead to positive experiences that facilitate improvement in resources because they provide protection by preventing resource loss or creating possibilities for resource replenishment [22, 24]. For the situation of students, participating in recovery activities to help experience recovery more frequently should lead to reduced stress as well as enhanced well-being. This association could be explained by the mediating effect of recovery experiences being involved in and resulting from these activities. However, this mediating effect has rarely been investigated by previous studies that based their analyses on student samples.

There is one particular study that examined the link between recovery activities and recovery experiences as well as associated outcomes in university students. Ragsdale et al. tested the importance of recovery experiences in the relationship between recovery activities and stress recovery during an exam period using a sample of 221 undergraduate students at an American university [25]. The authors examined whether recovery experiences pose the mediating mechanism between recovery activities and certain stress outcomes over a weekend (i.e., need for recovery and psychological strain). Recovery experiences were found to fully mediate the relationship between recovery activities and recovery quality. Ragsdale et al. argued that engaging in resource-providing recovery activities poses a promising way to not only experience recovery, but also to reduce academic stress [25]. In this study, however, recovery in students was only examined over a very short period of time. It would have been valuable to examine the role of recovery activities and recovery experiences over the course of several weeks or even months.

Overall, empirical findings are insufficient with regards to stress recovery in university students over longer periods of time. Exam preparation poses a particularly stressful study period for advanced law students that usually lasts more than one year [3, 11]. In this context, it would be desirable to better understand what might help students to reduce stress and enhance well-being over the course of exam preparation.

### 1.5 Rationale of the present study

The purpose of this study was to examine whether physical activity as a specific recovery activity would constitute a meaningful resource-providing activity for students during prolonged

exam preparation. Being confronted with high academic demands during exam preparation that threaten students' energetic and emotional resources, students are in need to restore those resources. In line with COR theory [22], we expected physical activity to have a stress-buffering potential that protects against loss of personal resources while facilitating resource gain. We therefore assumed that it would be beneficial when students invest additional resources (i.e., time for resource-providing activities) to help replenish the resources needed for recovery. We chose students' level of physical activity (i.e., reported hours spent with exercising per week) to be this resource-providing activity. Physical activity should help students (re)gain resources to facilitate stress recovery and have positive effects on their experienced stress and well-being at several measurement occasions.

The main goal of our study was to determine the extent to which recovery experiences mediate the relationship between physical activity (predictor) and stress as well as well-being (outcomes) over the course of prolonged exam preparation. We are not aware of any existing empirical studies that have examined the potential of physical activity and the role of recovery during prolonged periods of academic stress such as exam preparation. We deemed physical activity as one specific recovery activity to be a promising approach to help students better cope with academic strains during this stressful study period. Also, we thought physical activity to become more important for stress recovery over the course of exam preparation. On the basis of the literature and our expectations mentioned above, we derived two sets of directionally formulated hypotheses:

1. First set of hypotheses for study-related stress as an outcome:

   a. Students' recovery experiences mediate the negative association between physical activity and study-related stress at t1.

   b. Students' recovery experiences mediate the negative association between physical activity and study-related stress at t2.

   c. Students' recovery experiences mediate the negative association between physical activity and study-related stress at t3.

2. Second set of hypotheses for study-related well-being as an outcome:

   a. Students' recovery experiences mediate the positive association between physical activity and study-related well-being at t1.

   b. Students' recovery experiences mediate the positive association between physical activity and study-related well-being at t2.

   c. Students' recovery experiences mediate the positive association between physical activity and study-related well-being at t3.

## 2. Method

### 2.1 Sample and procedure

Our study comprised $N = 56$ participants (64% female) who were physically and mentally healthy. All subjects were advanced law students enrolled for at least six semesters and amidst their exam preparation to finish their law studies at one large German university. Most students were 22 years old ($M = 22.88$, $SD = 1.54$), in their 7th semester ($M = 7.57$, $SD = 0.95$, range 6–11), and in their third month of exam preparation ($M = 8.35$, $SD = 6.13$, range 1–24). The average student planned at least 18 months for exam preparation ($M = 15.27$, $SD = 3.08$,

range 10–18) and had about 13 months distance to taking their exams when the study began. About 90% of students were therefore at the beginning of their exam preparation. The typical period for exam preparation ranges anywhere between 12 and 24 months.

All students went through the longitudinal design of the study which entailed three measurement occasions over seven months (June 2018, November 2018, January 2019). Students needed to actively prepare for their final exams, take them within the next 18 months, and be motivated to participate across all measurement occasions to fit the inclusion criteria. We invited students to participate by advertising via the faculty's website, internal mailing lists, and social media. Data was collected in standardized terms by using the same paper-and-pencil questionnaire including the same measures each time. The time to complete one survey took about 20 min and informed consent was obtained. Students received a small financial compensation after the third measurement occasion. The entire study design was supervised and approved by the local ethics committee.

## 2.2 Measures

**2.2.1 Study-related stress.**   We measured students' levels of perceived stress as one of our primary outcomes by applying the Heidelberg Stress Index (HEI-STRESS) [36]. The HEI-STRESS is a brief scale that contains only three items to measure study-related stress and was specifically developed for the university context. For the first item, students rate their subjective stress on a scale of 0 (not at all stressed) to 100 (completely stressed out). Students then evaluated the frequency of general physical tension on a rating scale ranging from 0 (never) to 4 (daily). For the last item, students rate their level of stress in their life right now on a scale from 0 (not at all stressful) to 4 (very stressful). Students needed to answer with a two-week time reference. The final score of the brief stress scale ranges from 0 to 100 and is computed using the following formula (item 1 + (item 2 × 25) + (item 3 × 25)) /3. The internal consistencies were good at all measurement occasions ($\alpha$ = .82–.87).

**2.2.2 Study-related well-being.**   We assessed students' levels of subjective well-being as the other primary outcome besides stress. Again, we strived to apply an instrument that would fit the university context by measuring study-related well-being. For lack of an existing instrument and based on good experience from a previous study [11], we composed study-related well-being out of the following instruments considering both its cognitive and affective components.

*Life and study satisfaction*. We measured students' perceived study satisfaction using the Satisfaction With Life and Studies Scale (LSS) [37]. This questionnaire assessed the cognitive component of study-related well-being and is based on the Satisfaction With Life Scale [38]. The LSS was specifically developed for higher education settings and therefore includes study satisfaction as a subdomain of life satisfaction. Since students' academic and personal lives are two life domains that are thought to be closely related, both life and study satisfaction load on a single factor with $\alpha$ = .79 [39]. Seven items make up the full scale with four items for life and three items for study satisfaction. For the life satisfaction subscale, students evaluated their individual living circumstances in terms of their reported functioning and performance as well as their general level of life satisfaction (e.g., "How healthy and productive do you currently feel?", "How satisfied are you with your current life?"). For the study satisfaction subscale, students were asked to rate their performance and situational aspects of studying (e.g., "How satisfied are you with your current academic achievements?", "How satisfied are you with your current study situation?"). Students gave ratings for the last seven days on a rating scale ranging from 1 (not at all) to 5 (very much). The internal consistencies were satisfactory to good at all measurement occasions ($\alpha$ = .77–.84).

*Positive and negative affect.* We measured students' affect using the International Positive and Negative Affect Schedule Short Form (I-PANAS-SF) [40]. This questionnaire assessed the affective component of study-related well-being built on the Positive and Negative Affect Schedule (PANAS) [41]. The shortened PANAS comprised ten items for both subscales: five items for positive affect ("alert", "inspired", "determined", "attentive", "active") and five items for negative affect ("upset", "hostile", "ashamed", "nervous", "afraid"). Students were asked to give ratings on their experienced affective intensity for the last seven days on a rating scale ranging from 1 (never) to 5 (always). The internal consistencies for the combined scale of positive and negative affect (inverted) were mostly acceptable at all measurement occasions ($\alpha$ = .65–.80).

To get a composite score for study-related well-being, we integrated both instruments that assessed study-related satisfaction as well as positive and negative affect. We calculated an overall mean by combining the means of the LSS and the I-PANAS-SF (all items for negative affect were inverted) into one measure for study-related well-being. Joint internal consistencies were good at all measurement occasions ($\alpha$ = .81–.89).

**2.2.3 Recovery experiences.**  We assessed students' recovery experiences with the Recovery Experiences Questionnaire (REQ) [34]. The REQ measures the ability to recover from work-related demands during leisure (i.e., students' ability to unwind and recuperate from exam preparation). It contains four scales: psychological detachment (e.g., "I forget about work"), relaxation (e.g., "I use the time to relax"), mastery (e.g., "I seek out intellectual challenges"), and control (e.g., "I decide my own schedule"). We calculated the sum score of all 16 items to get a picture of students' subjective recovery status at each measurement occasion. Students evaluated their recovery experiences on a rating scale ranging from 1 (strongly disagree) to 5 (strongly agree). The internal consistencies were acceptable at all measurement occasions ($\alpha$ = .77–.78).

**2.2.4 Recovery activities.**  We focused on students' level of physical activity and exercise as a central recovery activity from academic coursework during exam preparation. Physical activity has repeatedly been reported to be a particularly important activity for psychological recovery in work-related contexts [26] and has been shown to be very effective in improving well-being [27]. Due to their high academic workload, we assumed that these effects would be transferable to students in exam preparation. We measured students' current level of physical activity by simply asking them to think of the last month and report the average hours spent with exercising per week at each measurement occasion.

## 2.3 Statistical analyses

Data were analyzed using SPSS (version 28). We used the PROCESS macro by Hayes [42] to test the mediation hypotheses, namely whether the relationship between physical activity and study-related stress as well as study-related well-being is mediated by recovery experiences. For each measurement occasion (t1–t3), we first computed basic models in which a direct effect of physical activity on the criterion (study-related stress or well-being) was calculated. Subsequently, two mediation models were calculated for each measurement occasion in which an indirect effect of physical activity over recovery experiences on both outcomes with respect to the respective criterion. One mediation model postulates an effect of physical activity (exogenous variable) on recovery experiences (mediator) as well as effects of the exogenous variable and the mediator on stress (criterion). The other mediation model includes an effect of physical activity (exogenous variable) on recovery experiences (mediator) as well as effects of the exogenous variable and the mediator on well-being (criterion). We computed confidence intervals (95%) to examine the significance of an indirect effect within the mediation models.

Each mediator was tested using the bootstrap method in the PROCESS tool with 1000 drawn samples. Mediation occurred when indirect effects significantly differed from zero in that the confidence intervals did not contain zero. Based on directionally formulated hypotheses, we applied one-sided testing.

## 3. Results

### 3.1 Descriptive statistics and intercorrelations

Means ($M$), standard deviations ($SD$) of all variables across the three measurement occasions are presented in Table 1. Students reported to do sports and exercise about $M = 3.54$ hours a week at t1 ($SD = 2.31$, range 0–14). This level of physical activity showed a further reduction from t2 ($M = 3.43$, $SD = 2.06$, range 0–9) to t3 ($M = 3.02$, $SD = 2$, range 0–10). Students spent less and less time engaging in recovery activities such as physical activity. This downward trend over the course of exam preparation is also reflected in the other variables. For recovery experiences, students reported an initial mean of $M = 3.44$ ($SD = 0.45$) at t1 which further declined from t2 ($M = 3.23$, $SD = 0.47$) to t3 ($M = 3.18$, $SD = 0.45$). Students reported their experienced recovery to decrease over time. There was a similar longitudinal development for both outcomes with some stabilization to be observed from t2 to t3. Moreover, students reported study-related well-being to decrease over time (SWB t1: $M = 3.30$, $SD = 0.5$) (SWB t2: $M = 3.12$, $SD = 0.6$) (SWB t3: $M = 3.13$, $SD = 0.61$). Vice versa, students reported study-related stress to increase across measurement occasions (SST t1: $M = 67.15$, $SD = 15.7$) (SST t2: $M = 75.85$, $SD = 15.7$) (SST t3: $M = 75.11$, $SD = 14.89$). See Table 1 for an overview on the descriptive results.

When looking at the intercorrelations between variables within each measurement occasion, there were some significant relationships among physical activity as one specific recovery activity, recovery experiences, and study-related stress and well-being as outcomes (see Table 2). Starting at t1, physical activity was correlated with well-being ($r = .31$, $p < .05$) but neither with recovery experiences nor stress. Also, recovery experiences were associated with well-being ($r = .33$, $p < .01$) but not with stress. At t2, physical activity was related with stress ($r = −.37$, $p < .01$) but did not correlate with well-being or recovery experiences. Moreover, recovery experiences were correlated with stress ($r = −.31$, $p < .05$) and well-being ($r = .50$, $p < .001$). At t3, physical activity was associated with stress ($r = −.36$, $p < .01$) and well-being ($r = .35$, $p < .05$) but not with recovery experiences. Furthermore, recovery experiences were associated with stress ($r = −.46$, $p < .001$) and well-being ($r = .57$, $p < .001$).

### 3.2 Mediation models

Our first set of hypotheses examined study-related stress as an outcome across all measurement occasions (t1–t3) (see Table 3 for results). Hypotheses 1a-c postulated that recovery

**Table 1. Means (M), standard deviations (SD), and Cronbach's alpha reliability coefficients (α) of all variables across measurement occasions.**

|  | t1 | | | t2 | | | t3 | | |
|---|---|---|---|---|---|---|---|---|---|
|  | *M* | *SD* | *α* | *M* | *SD* | *α* | *M* | *SD* | *α* |
| RAC | 3.54 | 2.31 | — | 3.43 | 2.06 | — | 3.02 | 2.00 | — |
| REX | 3.44 | .45 | .77 | 3.23 | .47 | .78 | 3.18 | .45 | .78 |
| SST | 67.15 | 15.70 | .87 | 75.85 | 15.70 | .88 | 75.11 | 14.89 | .82 |
| SWB | 3.30 | .50 | .81 | 3.12 | .60 | .87 | 3.13 | .61 | .89 |

*Notes*. *N* = 56. RAC = physical activity as specific recovery activity in hours per week, REX = recovery experiences, SST = study-related stress, SWB = study-related well-being.

**Table 2. Intercorrelations among all variables across measurement occasions.**

|        | RAC t2 | RAC t3 | REX t1 | REX t2 | REX t3 | SST t1 | SST t2 | SST t3 | SWB t1 | SWB t2 | SWB t3 |
|--------|--------|--------|--------|--------|--------|--------|--------|--------|--------|--------|--------|
| t1 RAC | .60*** | .60*** | .23 | .16 | .24 | −.19 | −.22 | −.24 | .31* | .26* | .25 |
| t2 RAC |  | .74*** | .09 | .16 | .35** | −.27* | −.37** | −.29* | .01 | .18 | .21 |
| t3 RAC |  |  | .21 | .21 | .26 | −.29* | −.26 | −.36** | .18 | .26 | .35* |
| t1 REX |  |  |  | .48*** | .21 | −.23 | −.12 | −.06 | .33** | .16 | .23 |
| t2 REX |  |  |  |  | .63*** | −.31* | −.31* | −.32* | .32* | .50*** | .51*** |
| t3 REX |  |  |  |  |  | −.40** | −.38*** | −.46*** | .21 | .42*** | .57*** |
| t1 SST |  |  |  |  |  |  | .71*** | .56*** | −.25 | −.44*** | −.38** |
| t2 SST |  |  |  |  |  |  |  | .73*** | −.24 | −.42*** | −.38*** |
| t3 SST |  |  |  |  |  |  |  |  | −.30* | −.50*** | −.58*** |
| t1 SWB |  |  |  |  |  |  |  |  |  | .61*** | .53*** |
| t2 SWB |  |  |  |  |  |  |  |  |  |  | .80*** |
| t3 SWB |  |  |  |  |  |  |  |  |  |  | — |

*Notes*. N = 56. RAC = physical activity as specific recovery activity, REX = recovery experiences, SST = study-related stress, SWB = study-related well-being.

*p < .05

**p < .01

***p < .001.

experiences would partially mediate the relationship between physical activity and study-related stress at t1–t3, respectively. In the first step, we computed a basic model in which physical activity predicted stress for all three measurement occasions. In the second step, we included recovery experiences as a mediator of the association between physical activity and stress. The results did not support our hypotheses 1a and 1b because there was no significant correlation between both recovery experiences and physical activity at t1 and physical activity and recovery experiences at t2. For this reason, no indirect effects from physical activity on stress were evident. These findings should be interpreted carefully due to the small sample size that renders even more pronounced changes in coefficients insignificant. However, the results did support our third hypothesis 1c in that recovery experiences partially mediated the relationship between physical activity and stress at t3. In this third model there was a significant

**Table 3. Results of mediation analyses for study-related stress across measurement occasions.**

|  |  | Standardized coefficients | | | Indirect effects[a] |
|------|-------|---------------------|-------------|-------------|------------------|
| Model | $R^2$ | RAC → SST | RAC → REX | REX → SST | RAC → SST |
| t1 Basic | .04 | −.19 |  |  |  |
| t1 Mediation | .07 | −.14 | .24* | −.20 | [−.12; .01] |
| t2 Basic | .14 | −.37** |  |  |  |
| t2 Mediation | .20 | −.32** | .17 | −.26* | [−.12; .03] |
| t3 Basic | .13 | −.36** |  |  |  |
| t3 Mediation | .33 | −.24* | .27* | −.46*** | [−.23; −.01] |

*Notes*. RAC = physical activity as specific recovery activity, REX = recovery experiences, SST = study-related stress. Basic model includes recovery activity as predictor only. Mediation model includes recovery activity as predictor and recovery experiences as mediator. → path weight

[a] = one-tailed bootstrapped confidence intervals

*p < .05

**p < .01

***p < .001.

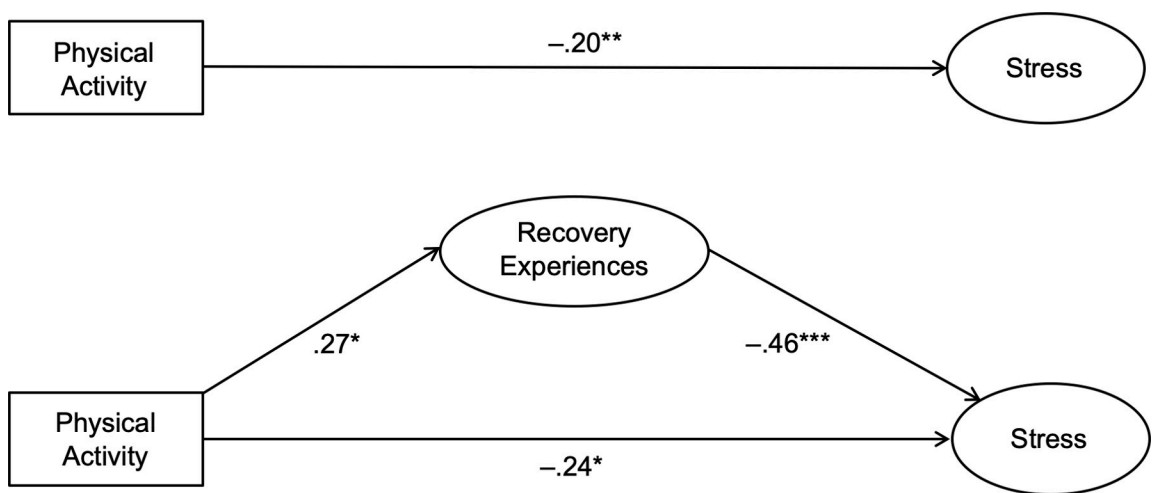

**Fig 1. Basic and mediational model for stress.** Depicted is the relationship between physical activity as specific recovery activity and study-related stress and its direct effects for the third measurement occasion (t3). Recovery experiences were included as mediator. $^*p < .05$; $^{**}p < .01$; $^{***}p < .001$.

indirect effect from physical activity on stress. Recovery experiences explained 33% of this relationship between physical activity and study-related stress as students progressed in time over their exam preparation. Fig 1 provides a graphical illustration of the statistical models with recovery experiences included as a mediator of this relationship at t3.

The second set of hypotheses looked at study-related well-being as an outcome across all measurement occasions (t1–t3) (see Table 4 for results). Hypotheses 2a-c postulated that recovery experiences would partially mediate the relationship between physical activity and study-related well-being at t1–t3, respectively. Again, we first computed a basic model in which physical activity predicted well-being. In the second step, we included recovery experiences as a mediator of the association between physical activity and well-being. The results of mediation analysis did support hypothesis 2a in that recovery experiences partially mediated the relationship between physical activity and well-being at t1. This was confirmed by a

**Table 4. Results of mediation analyses for study-related well-being across measurement occasions.**

| Model | $R^2$ | Standardized coefficients | | | Indirect effects[a] |
|---|---|---|---|---|---|
| | | RAC $\rightarrow$ SWB | RAC $\rightarrow$ REX | REX $\rightarrow$ SWB | RAC $\rightarrow$ SWB |
| t1 Basic | .10 | .31** | | | |
| t1 Mediation | .16 | .25* | .23* | .27* | [.01; .15] |
| t2 Basic | .03 | .18 | | | |
| t2 Mediation | .26 | .11 | .16 | .48*** | [−.04; .20] |
| t3 Basic | .12 | .35** | | | |
| t3 Mediation | .37 | .21* | .26* | .52*** | [.01; .26] |

*Notes.* RAC = Physical activity as specific recovery activity, REX = Recovery experiences, SWB = study-related well-being. Basic model includes recovery activity as predictor only. Mediation model includes recovery activity as predictor and recovery experiences as mediator. $\rightarrow$ path weight

[a] = one-tailed bootstrapped confidence intervals

$^*p < .05$

$^{**}p < .01$

$^{***}p < .001$.

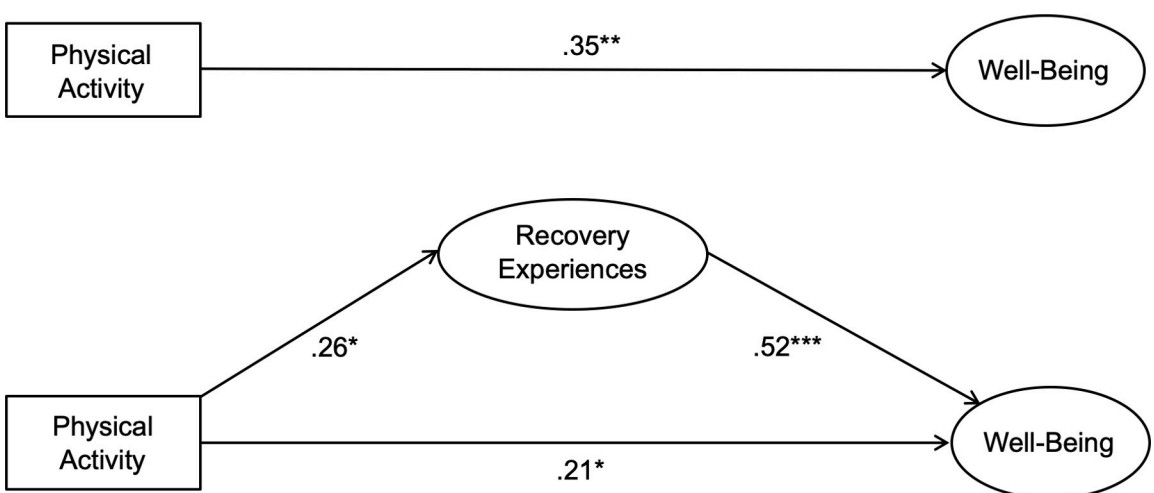

**Fig 2. Basic and mediational model for well-being.** Depicted is the relationship between physical activity as specific recovery activity and study-related well-being and its direct effects for the third measurement occasion (t3). Recovery experiences were included as mediator. $^*p < .05$; $^{**}p < .01$; $^{***}p < .001$.

significant indirect effect from physical activity on well-being. We found no support for hypothesis 2b when students had already advanced in their exam preparation at t2. There was no significant indirect effect from physical activity on well-being. Again, this finding should be interpreted with caution as to the small sample size. Nevertheless, results did provide support for hypothesis 2c because recovery experiences did partially mediate the relationship between physical activity and well-being at t3. This was also confirmed by a significant indirect effect from physical activity on well-being. The amount of variance explained by recovery experiences was 37% as students got further along with their exam preparation. Fig 2 shows a visual representation of the statistical models with recovery experiences included as a mediator of the aforementioned relationship at t3.

## 4. Discussion

The main purpose of this study was to determine the extent to which recovery experiences mediate the relationship between physical activity and study-related stress as well as well-being. Given the chronic stress characteristics of prolonged exam preparation [3], we examined whether physical activity as a specific recovery activity would constitute a meaningful resource-providing activity for advanced law students. We hypothesized that spending time on recovery activities with physical activity in particular results in reduced study-related stress levels and increased levels of well-being. We found partial support that this recovery process occurs through the mediating effect of recovery experiences and resource replenishment. This was in line with COR theory [22, 23] because students could invest additional resources (i.e., time to be physically active) to help replenish the resources needed for recovery in order to reduce stress and enhance well-being. Over the course of all three measurement occasions and in line with previous research [3], we saw a negative trend in all measures especially in recovery activity and recovery experiences. Students declined in their time spent with physical exercise and their experiences to feel recovered. There was a similar longitudinal pattern for study-related stress and well-being as outcomes. With regard to our mediation models, we found inconsistent results with recovery experiences being a mediator of the association between physical activity and stress as well as well-being at the respective measurement occasion.

## 4.1 Recovery experiences as a mediator of the relationship between physical activity and study-related stress

Our analyses revealed mixed results for recovery experiences mediating the negative association between physical activity and study-related stress across all three measurement occasions. Contrary to our first hypothesis (t1), mediation did not occur. We did not find a significant negative relationship between physical activity and stress, nor between recovery experiences and stress. The same was seen with regard to our second hypothesis (t2). Both physical activity and stress as well as recovery experiences and stress were negatively associated, but physical activity was unrelated to recovery experiences as students progressed with their exam preparation resulting in no mediation. However, in line with our third hypothesis (t3), we found a partial mediation effect for recovery experiences explaining the association between physical activity and stress with recovery experiences showing a much more pronounced effect on stress than physical activity. These findings suggest that recovery experiences play an important role in students' stress experience at later stages of their exam preparation. Recovery experiences showed stronger associations with stress across measurement occasions as reflected by the increasing magnitude of correlations. Moreover, the amount of variance explained by the statistical models that considered recovery experiences as a mediator kept increasing at each measurement occasion and was most pronounced at t3. However, it should also be noted at this point that we were only able to find comparatively low standard deviations for recovery experiences across all three measurement occasions. This could be a reason for weak correlations between the variables. Likewise, this could explain why we could only find a significant mediation effect of recovery experiences at t3, but not at t1 and t2.

It is somewhat surprising that physical activity was not correlated with stress at t1 because several studies have underlined the moderately positive association between exercising and mental health outcomes for work-related contexts [27] as well as academic settings [30, 31]. Findings are more inconsistent concerning the correlation between physical activity and perceived stress in college students [33, 43] while few researchers looked at academic examination periods. Our study examined students during exam preparation and this could explain why physical activity was less likely to have a positive impact on stress. Previous work showed that during exam periods, students reduced their time and frequency spent with exercising compared to non-exam conditions [5, 32]. At our first measurement occasion (t1), most students had to find their way in exam preparation which likely explains the lack of correlation between recovery-related variables and stress. Recovery does not seem to have played an overriding role with regard to students' subjective experience of stress at t1 and this relates to the non-occurring mediation effect of recovery experiences. Our second measurement occasion (t2) produced another pattern of correlations with only physical activity showing no significant relationship with recovery experiences which also resulted in a lack of mediation. The proposed mediation effect only occurred at our last measurement occasion (t3) and adding recovery experiences to the model showed a significant reduction in the original relationship between physical activity and stress.

Our findings can be put in context with one particular study that demonstrated recovery experiences to fully mediate between resource-providing recovery activities and recovery quality in a sample of college students during a short exam period [25]. Contrary to our study, Ragsdale et al. looked at recovery quality as a somewhat less rigorous outcome than perceived stress [25]. Recovery quality is much more related to recovery experiences than perceived stress which could explain why mediation only occurred at t3 in our data. However, study-related stress gave us a sense of how drained students felt and how they evaluated academic demands along with their ability to cope at each time point of our study (t1–t3). Another

reason why mediation did not also occur at t1 and t2 could be the small sample size of our study. Unlike Ragsdale et al., the small number of subjects within our statistical models may have caused the lack of mediation effects [25]. Furthermore, and unlike previous studies, the longitudinal approach of our study examined recovery over a longer period of time than just a weekend. The actual development of academic strain within exam preparation could have played a role in students' recovery activities and experiences at both t1 and t2. Because study-related stress is not perceived as so high earlier in students' exam preparation, recovery activities such as physical activity are not likely to have as great an effect on stress, nor are recovery experiences on stress (see correlations at t1). We think this to change the longer exam preparation lasts. Physical activity seems to become an important contrast to sitting down and studying (i.e., being physically inactive) every day. We assume that this contrast effect is psychological in nature, as recovery experiences not only showed a moderate negative correlation with stress, but were shown to partially mediate between physical activity and stress at later stages in exam preparation (t3).

Taken together, our results indicate recovery experiences gain importance in the relationship between physical activity and stress as students progress with their exam preparation. The prolonged academic strains of exam preparation lead to chronic stress [3] and therefore take a toll on students' cognitive and affective resources which translates into students having a greater need for recovery the further they get. Because recovery is a process that restores those resources that are in the process of deteriorating, recovery experiences seem to transmit the effects of physical activity into resource replenishment because they facilitate improvement in resources as described by COR theory [22, 24]. Physical activity as one specific recovery activity seems to offer a promising approach to help students better cope with academic strains during stressful exam preparation and this stress-buffering potential might unfold via feeling recovered after exercising.

## 4.2 Recovery experiences as a mediator of the relationship between physical activity and study-related well-being

Our results were somewhat more consistent for recovery experiences mediating the negative association between physical activity and study-related well-being across measurement occasions. In line with our first hypothesis (t1), there was a significant relationship among all variables which resulted in partial mediation. The same mediation effect occurred with regard to our third hypothesis (t3), but was absent for our second hypothesis (t2). On the second measurement occasion, we did not find a significant positive relationship between physical activity and well-being, nor between physical activity and recovery experiences. Students were not quite halfway through their exam preparation at this point. Taken together, these results indicate that recovery experiences make up an important part of students' subjective well-being during exam preparation. Similar to what was already seen with stress as an outcome, how students live through and experience their off-learning time appears to become more valuable the further they progress on the road of their exam preparation. Recovery experiences showed stronger associations with well-being across measurement occasions as reflected by the increasing magnitude of correlations. Furthermore, the amount of variance explained by the statistical models that considered recovery experiences as a mediator kept increasing at each measurement occasion. Again, it should be noted that we found rather low standard deviations for recovery experiences across all measurement occasions which could explain the lack of a mediation effect at t2.

Our findings were mostly consistent with previous studies showing that physical activity in general increases individual psychological well-being [27, 28]. Positive relationships between exercising and mental health outcomes are reported for both work-related contexts [27] and

higher education settings [29–31]. Going beyond those positive associations with well-being, we also found positive correlations between physical activity and feelings of recovery (i.e., recovery experiences) which is in line with one recent review [26]. The few studies that examined students during academic exam periods found reductions in both the duration and frequency of physical activity for those students who were facing exams compared to controls, but failed to look at their well-being [5, 32]. Our study extends previous findings by including study-related well-being as an outcome, which was also not addressed by Ragsdale et al. who found recovery experiences to be an important mediator of the association between resource-providing activities and recovery quality [25].

It is important to note, however, that the reported consistencies hold only for the first and third measurement occasions of our study (t1 and t3), but not for our second measurement occasion (t2). The proposed mediation effect did not occur at t2 because there were no positive correlations between physical activity and well-being, nor between physical activity and recovery experiences. This lack of a positive relationship between physical activity and recovery experiences resembles the correlational pattern we found for stress at t2 which leads us to assume that students were possibly confronted with a particular stressor. This idea is supported by looking at descriptive results because students reported their highest level of stress and their lowest level of well-being at t2 compared to the other measurement occasions. Going further than the small sample size of our study, we think that the reason to explain this irregularity may also lie in mock exams that are taken by students to train in solving legal cases. Mock exams represent an additional stressor during exam preparation and this stress may have had such a strong impact on students at t2 that negative effects on well-being and recovery experiences were evident. This explanation relates to a study that found cyclic motivational drops as students progressed with their exam preparation [9]. Therefore, lower levels of motivation might also have affected the relationship between both physical activity and well-being as well as recovery experiences.

Furthermore, our results suggest recovery experiences become more important to the relationship between physical activity and well-being as students progress with their exam preparation. Both students' cognitive and affective resources are depleted as a result of the prolonged nature of exam preparation and this makes recovery a key process to enhance study-related well-being. In line with COR theory, recovery experiences transmit the effects of leisure activities into resource replenishment [22]. Our findings support the fact that physical activity can be considered a promising way to help students maintain certain levels of well-being during stressful exam preparation. Importantly, it is not about physical activity alone, but students need to experience this recovery activity as actual psychological recovery from their everyday academic strains.

### 4.3 Limitations and further research

Despite its merits, our study has some shortcomings that need to be mentioned. First, we applied self-report measures using questionnaires only, which raised the concerns about social desirability of students' answers. Objective measures such as physiological assessments of stress recovery (e.g., cortisol levels or heart rate variability) would have ruled out such response biases by providing more precise data. However, we deemed self-reports to be a reasonable method to assess students' psychological experiences of recovery as well as study-related stress and well-being including reported time of physical activity per week. Second, even though we followed a longitudinal approach with three measurement occasions, our design does not allow for strong causal inferences. We did not control for variables that could have affected students' levels of physical activity at each measurement occasion (e.g., physical injuries, lack

of motivation). Third, in spite of carrying out our research at only one law school, the sample size was restricted to only a few students who engaged in our longitudinal study. Due to this small sample size, we cannot be very confident that a genuine mediation effect exists. Fourth, our study looked at a very specific group of students (i.e., advanced law students during exam preparation to pass their first state examination) which raises concerns about the generalizability of our results although this was the exact sample that we wanted to examine.

Beyond consideration of the above, future research should also examine other student outcomes such as exam grades as an indicator of academic achievement. It would be interesting to relate academic performance to stress recovery and examine whether those students who report higher levels of pre-exam recovery also achieve better grades. Moreover, further research is needed to better understand students' specific needs for stress recovery during prolonged study periods of academic pressure. For example, it is unclear whether students perceive exercising to be enjoyable and beneficial at all stages of exam preparation. There might be a shift towards engaging in more low-effort activities (e.g., watching TV) as exam preparation progresses. Furthermore, an experimental approach applying a recovery activity intervention would shed light on the actual effectiveness of physical activity for stress recovery. Future research should employ a randomized controlled trial to make causal statements about whether exercising can indeed be considered responsible for positive effects on student outcomes. Another aspect to be considered by further research is methodological improvements by using latent profile analyses that require bigger samples, for instance. This would allow for the examination of systematic patterns in students' recovery activities and experiences as well as to test whether these profiles differ in stress and well-being.

## 4.4 Practical implications

From a practical perspective, some interesting starting points can be derived from our study to help students better cope with prolonged and stressful study periods. This study sheds initial light on recovery activities and related experiences to reduce stress and enhance well-being during exam preparation. Our results suggest that physical activity as one specific recovery activity offers a feasible way to initiate the recovery process for advanced law students and that recovery experiences play a critical role for the positive effects to unfold. For example, physical activity allows for recovery experiences such as psychological detachment which promotes restoration and regeneration that reduces negative states (stress) and enhances positive states (well-being). To achieve recovery, sport and exercise belong to the most effective recovery activities [26] and this why students are recommended to find some time and become active during off-learning time. As outlined by previous studies, engaging in sports as one particular resource-providing activity poses a simple means of helping oneself and improving both physiological and psychological health [25, 27]. For example, low-threshold, easily accessible, and free university exercise classes could encourage law students to become more physically active during their studies, as suggested in a recent study [44]. In any case, higher education institutions are advised to support students in taking action by raising awareness about the supposed costs of prolonged academic strains. Because exam preparation is assumed to be effortful and resources are depleted to meet demands, recovery is needed to replenish those resources as described by COR theory [22]. Professors and academic staff at universities should make students more aware that long periods of learning also require specific actions to counteract the stress and regain resources. In doing so, they should refer to the scientific evidence and highlight physical activity as an effective way to improve physical and mental health.

More importantly, our findings point to recovery experiences as a mediating mechanism between activities and positive outcomes. To achieve recovery and regeneration through

physical activity, students need to take advantage of recovery experiences. For example, students could learn to become more aware of their respective mental state during off-learning time. This involves to consciously experiencing leisure as a pathway to recovery that fosters resource replenishment which could be achieved by giving students access to positive psychological interventions such as practicing mindfulness. In this context, students could also be invited to dwell on their experiences of feeling recovered and having recharged their batteries through journaling. Being a self-reflective form of writing about individual occurrences and experiences, journaling poses another promising method of positive psychology to reduce stress and enhance well-being. Universities would do well to establish suitable support formats that involve psychological advice to better cope with long-lasting study periods such as exam preparation [11]. Specifically, universities could assist students in making better use of their recreation experiences by creating curricula that intentionally set aside time for recovery activities and experiences. Reminding students to do something about their stress recovery would reduce the inhibition threshold and thus make the desired behavior more likely to translate into everyday practice. The fact that the process of recovery and the process of learning are directly linked and mutually dependent should not only be better understood by the universities, but also implemented accordingly.

## 5. Conclusion

Demanding study periods such as exam preparation put students through a period of long-lasting academic stress that also takes a toll on their well-being. Given the chronic stress characteristics and supposed decline in resources during prolonged exam preparation, recovery is an important mechanism to achieve resource replenishment. Physical activity is one specific recovery activity that helps students better cope with academic strains during such study periods. Our findings support the idea that recovery experiences are a key factor in the association between physical activity and study-related stress and well-being. We found evidence for recovery experiences to be a partial mediator of the relationship between physical activity and the outcomes. The findings indicate that the positive effects of recovery experiences become more sustained longitudinally meaning that how students live through and experience their off-learning time gets more valuable the further they progress with exam preparation. Recovery activities and experiences are both important and students can benefit from engaging in more physical activity and taking intentional advantage of recovery experiences.

## Author Contributions

**Conceptualization:** Tom Reschke.

**Data curation:** Tom Reschke.

**Formal analysis:** Tom Reschke, Katharina Reschke.

**Funding acquisition:** Tom Reschke.

**Investigation:** Tom Reschke.

**Methodology:** Tom Reschke, Katharina Reschke.

**Project administration:** Tom Reschke.

**Supervision:** Thomas Lobinger.

**Writing – original draft:** Tom Reschke.

**Writing – review & editing:** Katharina Reschke.

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
