## [Decision Letter · Decision Letter 0]

1 Apr 2024

PONE-D-23-39069Examining recovery experiences as a mediator between physical activity and study-related stress and well-being during prolonged exam preparation at universityPLOS ONE

Dear Dr. Reschke,

Thank you for submitting your manuscript to PLOS ONE. After careful consideration, we feel that it has merit but does not fully meet PLOS ONE’s publication criteria as it currently stands. Therefore, we invite you to submit a revised version of the manuscript that addresses the points raised during the review process.

Dear AuthorsThree reviewers read your manuscript and found it relevant and a good candidate for publication. At the same time, an update is required in the literature review section, note reviewer 1's comment on this topic. After updating literature based on articles published in the last two years (2023-2024), we will be happy to receive the updated version by resubmission and move forward.

We look forward to receiving your revised manuscript.

Kind regards,

Gal Harpaz, Ph.D.

Academic Editor

PLOS ONE

Journal Requirements:

2. PLOS requires an ORCID iD for the corresponding author in Editorial Manager on papers submitted after December 6th, 2016. Please ensure that you have an ORCID iD and that it is validated in Editorial Manager. To do this, go to ‘Update my Information’ (in the upper left-hand corner of the main menu), and click on the Fetch/Validate link next to the ORCID field. This will take you to the ORCID site and allow you to create a new iD or authenticate a pre-existing iD in Editorial Manager. Please see the following video for instructions on linking an ORCID iD to your Editorial Manager account: https://www.youtube.com/watch?v=_xcclfuvtxQ.

Reviewers' comments:

Reviewer's Responses to Questions

**Comments to the Author**

1. Is the manuscript technically sound, and do the data support the conclusions?

Reviewer #1: Yes

Reviewer #2: Yes

Reviewer #3: Yes

2. Has the statistical analysis been performed appropriately and rigorously? 

Reviewer #1: Yes

Reviewer #2: Yes

Reviewer #3: Yes

3. Have the authors made all data underlying the findings in their manuscript fully available?

Reviewer #1: No

Reviewer #2: Yes

Reviewer #3: Yes

4. Is the manuscript presented in an intelligible fashion and written in standard English?

Reviewer #1: Yes

Reviewer #2: Yes

Reviewer #3: Yes

5. Review Comments to the Author

Reviewer #1: REVIEWER’S REPORT

Summary of Manuscript Number: PONE-D-23-39069

• The authors conducted a study on “examining recovery experiences as a mediator between physical activity and study-related stress and well-being during prolonged exam preparation at university.”

• The authors applied a longitudinal design and approached students at three measurement occasions over seven months.

• The participants were 56 advanced law students enrolled at one large German university.

• The findings showed a negative trend in recovery-related variables and the outcomes as exam preparation progressed.

• The authors realized that their results suggest that the positive effects of recovery experiences related to physical activity become more sustained as exam preparation progresses and have a particularly positive impact on well-being.

Soundness and Quality of the Paper

The main claim of the paper thus “the role of recovery experiences as a mediator of the relationship between physical activity as one specific recovery activity and both study-related stress and well-being” is significant for the discipline. The authors’ claim was properly placed in the context of previous literature which they treated fairly. The data and the analyses fully supported their claim. The methods applied are appropriate. The manuscript is well organized and written clearly enough. I believe it will be accessible to non-specialists.

Comments to the Author(s)

The authors should reconsider their literature review. In all, they cited 63 articles, only 23 (36.5%) were published between 2014 to 2023. The remaining 40 (63.5%) articles were too old for this kind of article. Therefore, I suggest that whatsoever literature they want to use, it should not be older than 2014. If this is addressed, then, the paper can be considered for publication.

Reviewer #2: Comments for the author:

This study provides very important statistical data and descriptive results on physical activity (sport, exercise) as a recovery activity would constitute a meaningful resource to reduce chronic stress and to increase well- being, as well as a good mental health for law students during their academic or school life (specially in exams).

The abstract is good. But introduction is very long or extensive. Results, discussion, and conclusion are good. References: All references are cited in manuscript.

Reviewer #3: Scholarly display of great command over the subject matter. Congratulations on a paper well written. All key sections of the manuscript have been adequately written consistent with the guidelines of the journal.

6. PLOS authors have the option to publish the peer review history of their article (what does this mean?). If published, this will include your full peer review and any attached files.

Reviewer #1: No

Reviewer #2: No

Reviewer #3: **Yes: **BOTHA Nkosi Nkosi

---

## [Author Response · Author response to Decision Letter 0]

21 Apr 2024

Reply to Editor 

Before we provide a detailed report of the changes we have made to the manuscript, we want to thank you for your valuable and much appreciated feedback and for the many helpful suggestions on how to improve our paper. We have addressed all of concerns that were raised and followed the advice. 

We think that the revision process has substantially improved the manuscript and we hope that you will deem the revised manuscript suitable for publication in PLOS ONE. 

Sincerely, 

The Authors 

Reply to Reviewer 1

The authors conducted a study on “examining recovery experiences as a mediator between physical activity and study-related stress and well-being during prolonged exam preparation at university.” The authors applied a longitudinal design and approached students at three measurement occasions over seven months. The participants were 56 advanced law students enrolled at one large German university. The findings showed a negative trend in recovery-related variables and the outcomes as exam preparation progressed. The authors realized that their results suggest that the positive effects of recovery experiences related to physical activity become more sustained as exam preparation progresses and have a particularly positive impact on well-being.

Soundness and Quality of the Paper

The main claim of the paper thus “the role of recovery experiences as a mediator of the relationship between physical activity as one specific recovery activity and both study-related stress and well-being” is significant for the discipline. The authors’ claim was properly placed in the context of previous literature which they treated fairly. The data and the analyses fully supported their claim. The methods applied are appropriate. The manuscript is well organized and written clearly enough. I believe it will be accessible to non-specialists.

Comments to the Author(s)

The authors should reconsider their literature review. In all, they cited 63 articles, only 23 (36.5%) were published between 2014 to 2023. The remaining 40 (63.5%) articles were too old for this kind of article. Therefore, I suggest that whatsoever literature they want to use, it should not be older than 2014. If this is addressed, then, the paper can be considered for publication.

Thank you for this good comment. We have deleted many references that are older than 2014 from the manuscript. This effort now results in a significantly better ratio between older and newer studies (44 cited articles in total with approx. 40% older studies and 60% newer studies). However, we think that some should remain in the paper for the following reasons (considered in the order in which they appear in the manuscript):

• The study by Oaten and Cheng (2005) is a fundamental study that has influenced subsequent studies in the field. It is typically cited in other recent research that has examined stress in students and the role of physical activity (e.g., Lines et al., 2021).

• The study by Glöckner et al. (2013) enables interested readers who would like more background information to find out more about studying law in Germany and the examination modalities. To our knowledge, there is no other study that goes into this in such detail.

• The definitions of stress and well-being represent the central outcomes of our study, which is why we cite the work of Lazarus and Folkman (1984) and Diener (1984), who defined these constructs within psychological literature and whose wording is still agreed upon in the scientific community today. Furthermore, it also helps the reader to retrace the constructs origins.

• The Conservation of Resources Theory (COR; Hobfoll, 1989, 2002) is one of the predominant theoretical approaches for understanding the recovery process. As this model is used to derive the research question, we felt it necessary to mention the original work on this model.

• Because the research situation in favor of physical activity with university students is less detailed, especially during final examinations, we would like to leave the cited studies in the manuscript (Griffin et al., 1993; Molina-García et al., 2011; Oaten & Cheng, 2005; Tyson et al., 2010; Steptoe et al., 1996). This serves to clarify the added value of our study against the background of the research gap.

• The study by Sonnentag and Fritz (2007) developed an initial taxonomy of recovery experiences comprising psychological detachment from work, relaxation, mastery, and control. This is a key study that is commonly cited when describing the individual components of recovery experiences. This study also forms the basis for the Recovery Experience Questionnaire (REQ), which we describe later in the measures section.

• The study from Ragsdale et al. (2011) is described in more detail in the manuscript because it is considered an important preliminary work for the present study (recovery processes were also examined in more detail in mediation analyses for the first time in an academic context).

• Both studies Holm-Hadulla and Hofmann (2007) and Holm-Hadulla et al. (2009) are important because they developed our measurement tool for study-related well-being. Both studies are in turn based on the work of Diener et al. (1985), which we would like to make clear at this point. The work of Thompson (2007), which is based on the preliminary work of Watson et al. (1988), is similar.

• We cite the work of Hayes (2013) to illustrate the statistical analysis method we used and exactly how we went about it.

Reply to Reviewer 2

This study provides very important statistical data and descriptive results on physical activity (sport, exercise) as a recovery activity would constitute a meaningful resource to reduce chronic stress and to increase well- being, as well as a good mental health for law students during their academic or school life (specially in exams).

The abstract is good. But introduction is very long or extensive. Results, discussion, and conclusion are good. References: All references are cited in manuscript.

Thank you for this valuable comment. We have now shortened the introduction significantly to improve the reading flow.

Reply to Reviewer 3

Scholarly display of great command over the subject matter. Congratulations on a paper well written. All key sections of the manuscript have been adequately written consistent with the guidelines of the journal.

Thank you for your kind words.

---

## [Editor Report · Decision Letter 1]

25 Jun 2024

Examining recovery experiences as a mediator between physical activity and study-related stress and well-being during prolonged exam preparation at university

PONE-D-23-39069R1

Dear Dr. Reschke,

We’re pleased to inform you that your manuscript has been judged scientifically suitable for publication and will be formally accepted for publication once it meets all outstanding technical requirements.

Kind regards,

Gal Harpaz, Ph.D.

Academic Editor

PLOS ONE

Additional Editor Comments (optional):

Dear Authors

Much appreciation for the efforts and thoroughness with which you addressed the reviewers' comments. The revised version of the article deserves to be published, the changes you made in it provide an excellent response to the comments and suggestions of the reviewers.

Therefore, the article is accepted for publication in its current version.

Best regards

Dr. Gal Harpaz
---

## [Editor Report · Acceptance letter]

10 Jul 2024

PONE-D-23-39069R1 

PLOS ONE

Dear Dr. Reschke, 

I'm pleased to inform you that your manuscript has been deemed suitable for publication in PLOS ONE. Congratulations! Your manuscript is now being handed over to our production team.

Kind regards, 

on behalf of

Dr. Gal Harpaz 

Academic Editor

PLOS ONE